# Tolerability of oral itraconazole and voriconazole for the treatment of chronic pulmonary aspergillosis: A systematic review and meta-analysis

Ronald Olum[1], Joseph Baruch Baluku[2,3], Andrew Kazibwe[4,5], Laura Russell[6], Felix Bongomin[5,7]*

1 School of Medicine, College of Health Sciences, Makerere University, Kampala, Uganda, 2 Department of Programs, MildMay Uganda, Wakiso, Uganda, 3 Division of Pulmonology, Mulago National Referral Hospital, Kampala, Uganda, 4 The AIDS Support Organisation, Kampala, Uganda, 5 Department of Medicine, School of Medicine, College of Health Sciences, Makerere University, Kampala, Uganda, 6 Medical Library, Manchester University NHS Foundation Trust, Manchester, United Kingdom, 7 Department of Medical Microbiology and Immunology, Faculty of Medicine, Gulu University, Gulu, Uganda

* drbongomin@gmail.com

**Data Availability Statement:** All relevant data are within the manuscript and its Supporting Information files.

**Funding:** The author(s) received no specific funding for this work.

## Abstract

### Background

Chronic pulmonary aspergillosis (CPA) requires prolonged treatment with itraconazole or voriconazole. However, adverse events (AEs) are common with the use of these agents, with the need to discontinue the offending drug in a significant proportion of the patients. The aim of this study was to evaluate the frequency of adverse events of itraconazole and voriconazole for the treatment of CPA.

### Methods

We searched Embase and Medline to select clinical studies providing information on AEs to itraconazole or voriconazole for the treatment of CPA from inception to May 2020. Reviews, single case reports, and case series reporting less than 10 patients were excluded. Random effect meta-analysis was performed using STATA 16.0.

### Results

We included 9 eligible studies with an overall total of 534 CPA patients enrolled. Of these, 69% (n = 366) were treated with voriconazole and 31% (n = 168) with itraconazole. The median daily dose of both itraconazole and voriconazole used was 400mg. In a pooled analysis, AEs were observed in 36% (95% CI: 20–52%, N = 366) of patients on voriconazole and 25% (95% CI: 18 to 31%, N = 168) in those treated with itraconazole. Discontinuation rate due to AEs was the same for both drugs; 35% (47/366) and 35% (15/168) for voriconazole and itraconazole, respectively. There were 70 AEs reported with itraconazole use, the commonest being cardiotoxicity (29%). Skin AEs (28%) were the most frequent among the 204 AEs reported with voriconazole use. None of the studies compared the tolerability of itraconazole head-to-head with voriconazole.

**Competing interests:** The authors have declared that no competing interests exist.

## Conclusions

AEs due itraconazole and voriconazole are common and may lead to discontinuation of treatment in a significant proportion of patients. This information can be used to educate patients prior to commencement of these antifungal therapies.

## PROSPERO registration number

CRD42020191627

## Introduction

Chronic pulmonary aspergillosis (CPA) is a progressive lung disease characterised by cavitary lesions with evidence of paracavitary infiltrates, new cavity formation, or expansion of cavity size over time with or without a fungal ball accompanied by chronic pulmonary or systemic symptoms of 3 months duration or more, including at least 1 of the following symptoms: weight loss, productive cough, or hemoptysis [1]. CPA remains a global public health problem with a substantial morbidity and mortality. CPA occurs in patients with underlying structural lung diseases, notably healed pulmonary tuberculosis and chronic obstructive pulmonary diseases [2].

Management of CPA is dependent on the radiological phenotype. Simple aspergilloma requires surgical intervention; however, chronic cavitary pulmonary aspergillosis, which may progress to chronic fibrosing pulmonary aspergillosis, generally requires medical treatment with antifungal agents [3, 4].

Long-term oral antifungal therapy is the cornerstone of management of CPA and is associated with improved quality of life and reduced disease activity [5, 6]. Itraconazole and voriconazole are interchangeably used as first-line agents for the management of CPA with response ranging anywhere between 30 and 90% [4, 7–12]. Adverse events (AE) limit the use of these first line agents in a significant proportion of patients, requiring discontinuation in up to 30% of cases [4]. The precise predictors of these adverse events remain unknown. Rates of AEs may be affected by both the patient's demographics and co-morbidities [13].

Few studies have reported adverse events to itraconazole and voriconazole from routine use and in clinical trial settings [14]. To-date, no systematic review has been undertaken to comprehensively evaluate AEs in this group of patients.

Therefore, this systematic review and meta-analysis was performed to evaluate the rates of AEs among patients being managed for CPA with itraconazole or voriconazole.

## Methods

### Search strategy

This study was performed according to the Preferred Reporting Items for Systematic Reviews and Meta-Analyses (PRISMA) statement [15]. The PRISMA checklist is available in S1 File. The protocol for conducting this systematic review is registered in the PROSPERO database (registration number: CRD42020191627)

We searched all studies published from inception to May 2020 from EMBASE and Medline databases.

The following search terms were used: "chronic pulmonary aspergillosis," "CPA," "*Aspergillus* nodule," "simple aspergilloma," "chronic cavitary pulmonary aspergillosis," "CCPA,"

"chronic fibrosing pulmonary aspergillosis," "CFPA," "itraconazole," "voriconazole," "adverse events," "side-effects," and "toxicities".

In addition, manual literature search of all the references of the included articles was performed to find relevant studies. The search was limited to human studies and only studies written in English were selected.

The search strategy is available as S2 File.

### Study selection criteria

The studies found through databases that were duplicates were removed using the HDAS program (National Institute for Health and Care Excellence, London, United Kingdom). Records were initially screened by title and abstract by two independent reviewers (F.B and R.O) to exclude those not related to the current study. The full text of potentially eligible records was retrieved and examined. Any discrepancies were resolved by consensus.

The PRISMA criteria for searching and selecting studies were used: The following inclusion criteria were applied to identify eligible studies: (i) published in the English language from inception to May 2020; (ii) designed as retrospective or prospective observational study or randomized clinical trial; (iii) reporting adverse events to itraconazole, voriconazole or both during treatment of CPA (iv) providing the incidence of toxicity and whether antifungal therapy was discontinued or not. We excluded studies on other forms of aspergillosis and single case reports and case series reporting less than 10 cases.

### Data extraction

The following items were extracted from each article: year of publication, authors, study design, antifungal agent administered, sample size, age, underlying diseases, dose and duration of administration, and safety outcomes including tolerability and the specific AEs. Two reviewers (R.O and F.B) independently extracted the data, and differences were resolved by consensus.

### Study outcomes

The study outcomes were the cumulative incidence of AEs and the proportion of patients who discontinued antifungal therapy due to adverse events.

### Analysis

A random-effects meta-analysis was performed using *metaprop* command [16] in STATA 16.0. Heterogeneity across studies was assessed using the $I^2$ index. Publication bias was assessed using a funnel plot and sensitivity analysis performed. $P<0.05$ was considered statistically significant at the 95% confidence interval. Descriptive statistics were used to summarise individual patient data. A narrative synthesis was also used to present the results and discussion of the study.

## Results

### Characteristics of studies

A total of 9 studies [4, 5, 11, 17–22] were included in the systematic review and 8 studies were eligible for meta-analysis, **Fig 1**. **Table 1** summarizes the characteristics of studies included in the review and meta-analysis. Of the 10 studies, 6 were retrospective [4, 5, 11, 17, 18, 22], 1 randomised clinical trial [19] and 2 prospective, multicentre studies [20, 21]. Majority of the studies were from the United Kingdom (UK, n = 4) [4, 5, 11, 17] and France (n = 2) [20, 22]. Overall, the studies had 534 patients with mean/median age ranging from 36 to 66 years.

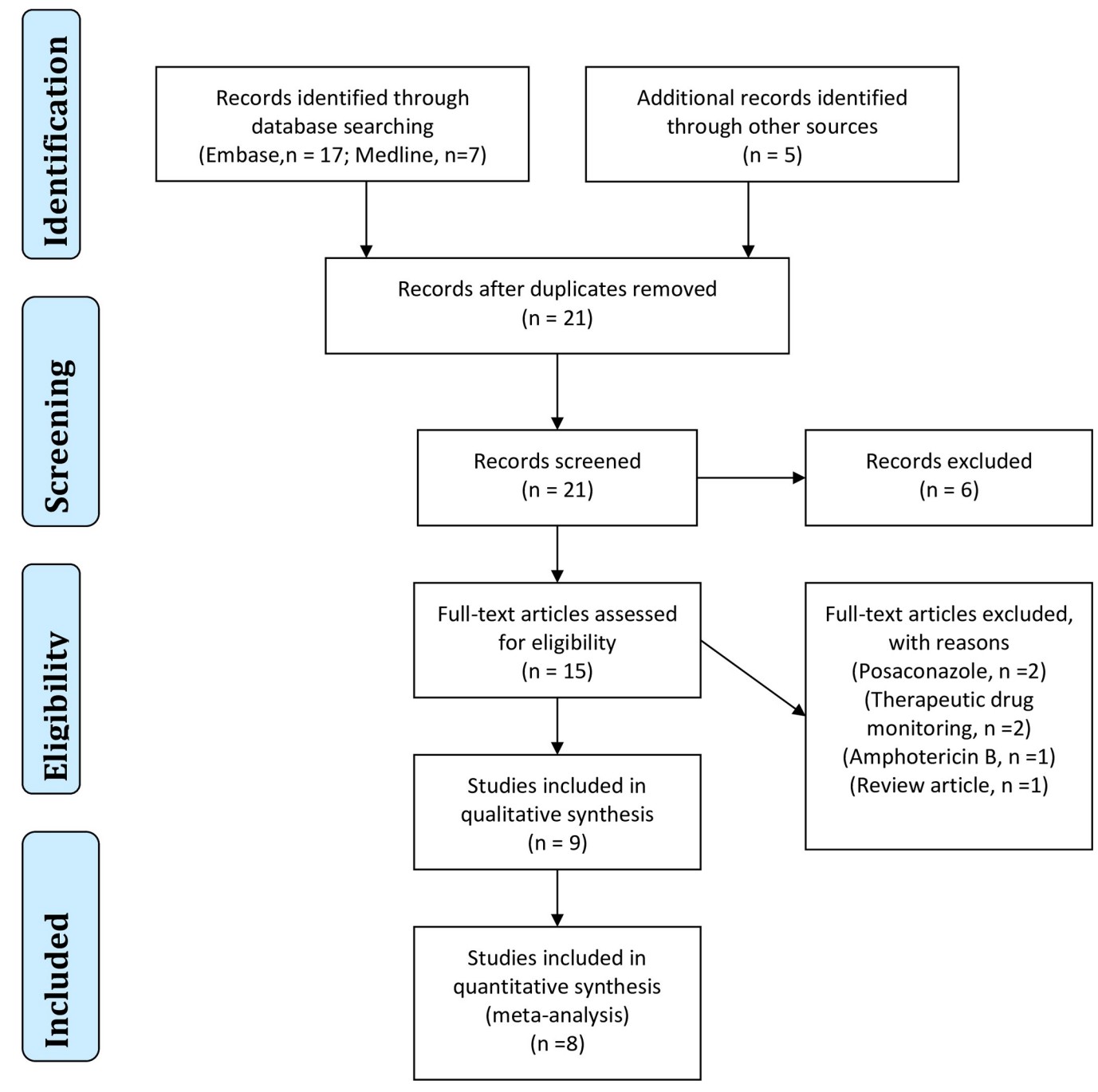

**Fig 1. PRISMA flow diagram showing study selection.**

Patients in 8 studies [4, 5, 11, 17, 18, 20–22]were treated with voriconazole (n = 366, 69%) and with itraconazole (n = 168, 31%) in two studies [4, 19], **Table 1**.

### Adverse events

**Voriconazole.**    AEs to voriconazole was analysed in 8 studies. AE incidence rates ranged from 13% to 86%, Table 1. Overall, pooled incidence rates of AE in patients who received vori-conazole for CPA treatment was 36% (95% CI: 20–52%; $I^2$ = 91.94%, $P$<. 001), **Fig 2**.

**Table 1. Characteristics of the eligible studies.**

| Study (Year) | Study Type | Country | Patients (N = 534) | Age (Median or Mean) | Intervention | Dosage | AE (n) | Discontinued: n (%) |
|---|---|---|---|---|---|---|---|---|
| Bongomin (2019) [17] | Retrospective Observational | UK | 21 | 66 | Voriconazole | 300-500mg/day, Mean: 400mg/day | 18 | 10 (56) |
| Bongomin (2018) [4] | Retrospective Observational | UK | 151 | 65 | Itraconazole | 100-600mg/day, Median: 400mg/day | 35 | 15 (43) |
| Bongomin (2018) [4] | Retrospective Observational | UK | 43 | 65 | Voriconazole | 200-500mg/day; Median: 400mg/day | 10 | 5 (50) |
| Cucchetto (2015) [18] | Retrospective Observational | Italy | 21 | 52 | Voriconazole | LD: 800mg, then 400mg/day | 6 | 6 (100) |
| Al-Shair (2013) [5] | Prospective Longitudinal | UK | 122 | 59 | Voriconazole |  | 46 | 9 (20) |
| Agarwal (2013) [19] | Open-labelled, Randomised Control Trial | India | 17 | 36 | Itraconazole | 400mg/day for 6 months | 8 | 0 (0) |
| Cadranel (2012) [20] | Prospective Non-Comparative Multicentre study | France | 48 | 58 | Voriconazole | 100-200mg twice daily | 7 | 7 (100) |
| Saito (2012) [21] | Prospective Non-Comparative Multicentre study | Japan | 71 | 66 | Voriconazole | 6mg/kg twice daily | 39 | 2 (5) |
| Camuset (2007) [22] | Retrospective Observational | France | 24 | 59 | Voriconazole | - | 3 | 3 (100) |
| Jain (2005) [11] | Retrospective Observational | UK | 16 | - | Voriconazole | 150-200mg twice daily | 5 | 5 (100) |

**Abbreviations: UK- United Kingdom; AE- Adverse Event; LD- Loading Dose.**

A funnel plot was generated to show the distribution of the included studies, S1 Fig. A sensitivity analysis was then performed using 3 studies that were under the funnel [5, 11, 18]. Pooled proportion of patients with AE from these studies was 36% (95%CI: 28% - 43%, $I^2$: 0.00%, $P$ = .64). The forest plot for the sensitivity analysis is available as S2 Fig.

All studies except Camuset et al [22] reported frequencies of individual AEs that were observed in patients. A total of 204 AEs was reported. The commonest AEs were skin toxicity

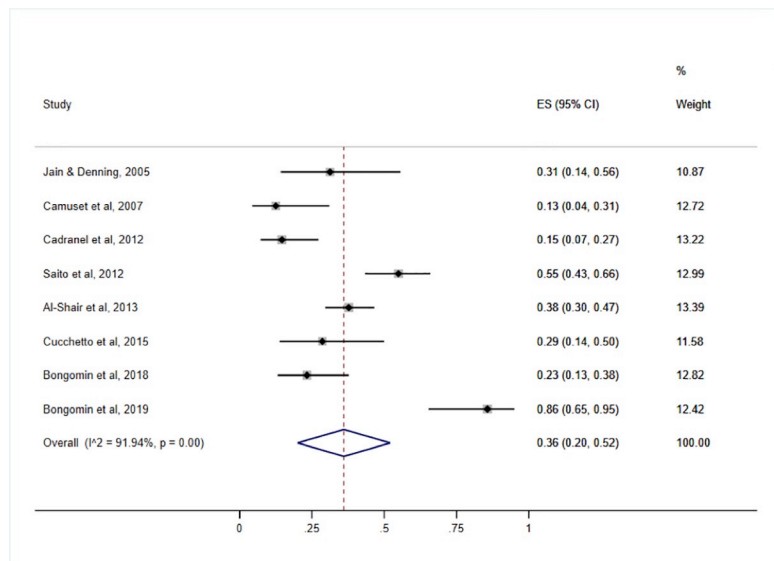

**Fig 2. Pooled proportion of adverse events in the studies included (N = 8).**

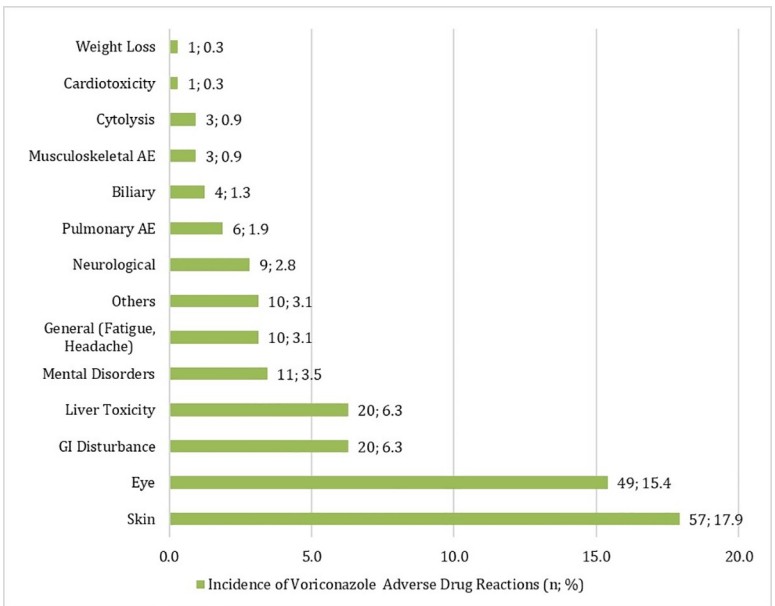

**Fig 3. Adverse events to voriconazole.**

(n = 57, 27.9%), visual disturbances (n = 49, 24.0%), hepatotoxicity (n = 20, 9.8%), gastrointestinal disturbances (n = 20, 9.8%) and mental disorders (n = 11, 5.4%), **Fig 3**.

**Itraconazole.** Only two studies reported AE following oral itraconazole use in patients with CPA [4, 19]. In one study [19], AE incidence rate was 47% and 23% in a UK study [4]. Overall pooled incidence rate of AE following oral itraconazole from the two studies was 25% (95% CI: 18–31%; $I^2$ = 0.00% p<. 001). A total of 70 AEs was reported. The most common AEs were cardiotoxicities (n = 20, 28.6%), gastrointestinal disturbance (n = 17, 24.3%) and peripheral neuropathy (n = 10, 14.3%), **Fig 4**.

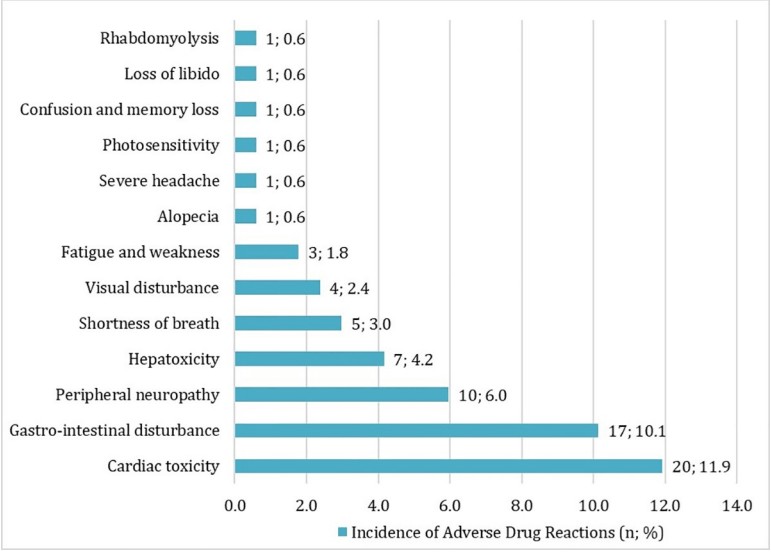

**Fig 4. Adverse events to itraconazole.**

## Discontinuation from therapy

Table 1 summarizes the proportion of patients who discontinued voriconazole or itraconazole due to AEs. Of 134 patients who experienced AEs to voriconazole, 35% (n = 47) patients were discontinued. Discontinuation rates ranged from 5 to 100% of those who had AEs across studies. Overall, 35% (n = 15) of the 43 patients treated with itraconazole who developed AEs discontinued treatment.

## Discussion

CPA is a debilitating progressive disease that highly affects the quality of life of the patients. Antifungal therapies have shown to improve symptoms and quality of life in up to two-thirds of patients [5, 23]. Itraconazole and voriconazole are used interchangeably as first-line agents for the long-term management of CPA [3, 6, 8]. However, voriconazole is preferred for patients with more severe disease or those with intra-cavitary fungal balls [4]. The reported efficacy of itraconazole and voriconazole varies from study-to-study, with a range of 30–93% and 50-67%, respectively [19]. Intravenous agents like amphotericin B and echinocandins have also shown comparable response rates in the treatment of CPA [24]. In a trial where CPA patients were initially stabilised with intravenous antifungals, oral voriconazole maintenance therapy showed better effectiveness than oral itraconazole for clinical improvement in CPA patients [7].

Posaconazole and isavuconazole are alternative agents for the management of CPA in patients who may not tolerate or have developed resistance to first-line drugs [17, 25, 26]. However, multiple adverse drug reactions have been reported with the use of triazoles, ranging from mild AE like nausea and vomiting to severe life-threatening AE like hepatotoxicity and cardiotoxicity.

In the present study, we synthesize the incidence of AEs following oral itraconazole and voriconazole reported in available studies. Our study shows that patients may better tolerate oral itraconazole than voriconazole in the treatment of CPA. The discontinuation rates due to AE are comparable across studies. However, a randomised clinical trial is required to confirm these findings.

AEs are commonly observed in patients on itraconazole. Transient elevation of liver enzymes (1.5 to 2 times) was observed in two patients in the first 2-3 months of treatment. In this study, most of the AE were mild and none required discontinuation from therapy [19]. Of the 23.1% (n = 35) of the patients who experienced AE in another UK retrospective, up to 37% were switched to a different azole (voriconazole and posaconazole) and 43% (n = 15) required discontinuation of therapy over a 12-month period of therapy [4]. The most common AE reported include heart failure as a result of cardiac toxicity, gastrointestinal disturbances and peripheral neuropathies. Life threatening AE like rhabdomyolysis have also been reported [4].

We observed AEs to voriconazole ranging from 12.5% to 85.7% among individual studies. In a recent study, up to 86% of the patients who received a median dose of 400mg/day of voriconazole experienced at least one AE, 56% of the patients had to be discontinued from therapy [17]. In our study, voriconazole-related skin toxicity, ocular toxicity and gastrointestinal disturbances were the most frequently observed AE which is in line with other studies reporting voriconazole related AEs [27]. Photosensitivity leading to erythematous rashes were reported in up to 6 studies [4, 5, 11, 18, 20, 22] but were prevented by use of sunscreens. Voriconazole-induced photosensitivity reactions include erythematous eruptions like facial erythema, cheilitis, skin hyperpigmentation, exfoliative dermatitis, discoid lupus erythematous, Steven-Johnson Syndrome, toxic epidermal necrolysis [28, 29]. Skin toxicities may lead to a melanoma and non-melanoma skin cancers, which calls for pharmacovigilance especially when initiating immunocompromised patients on voriconazole [30].

Patients also experienced visual disturbances like blurred vision, retinal flashes and photophobia; however, no study has reported blindness as a result of voriconazole use. Animal studies have demonstrated that voriconazole ocular toxicities could be due to the blockage of TRPM1 cation channels in retinal ON-bipolar cells [31]. Up to 83% of patients in a double-blind randomised controlled study suffered at least one visual AE following voriconazole therapy [32]. Other neurological conditions such as paraesthesia [11], vertigo [20] were also observed. Hepatotoxicity ranging from mild to severe was frequently reported in studies. Grade 3 liver toxicity was described in 2 patients, which led to discontinuation of therapy [20]. Derangement of liver enzymes was also observed in all the 8 studies that reported voriconazole use for CPA. Psychological symptoms like hallucination and memory disorders were also reported in some studies [18, 21]. The gastrointestinal AEs were mostly mild; included loss of appetite, nausea, vomiting and diarrhoea and did not necessitate discontinuation from treatment. Other AEs like QT prolongation, cytolysis, cholestasis, insomnia and general symptoms like headache were also observed.

We observed a very high rate of discontinuation of itraconazole (35%) or voriconazole (35%) due to AEs. This presents an issue as such patients are at risk of deterioration in their quality of life or relapse of CPA. Recently, we showed that over 40% of CPA patients who discontinue antifungals do relapse within the first 6 months [33]. Patients with bilateral diseases and those who had had short duration of antifungal treatment before discontinuing treatment were more likely to relapse.

Itraconazole and voriconazole have variable pharmacokinetics with significant drug-drug and drug-food interactions [34]. This results into varied serum drug levels and tissue exposures among patients. In most cases, AEs to these agents are dose dependent and therefore therapeutic drug monitoring should be a standard of care for patients on long-term treatment to optimise care and minimise AEs [34].

Our study however faces some limitations. The heterogeneity of the study has to be taken into consideration while interpreting the results of this study. Most of the patients in this study are from diverse ethnical origin, co-morbidities, different drug dosages and duration of therapy. Some studies also had incomplete reports of AE and were not taken into consideration during the analysis. However, results from a less heterogeneous group in a sensitivity analysis were consistent with the overall results.

## Conclusion

From this study, treatment related AE occurred in between 20 and 52% of patients commenced on voriconazole and between 18% and 31% of patients commenced on itraconazole. Our study suggests that oral itraconazole may be a more tolerable agent compared to voriconazole for the treatment of CPA. However, to date no study has directly compared the tolerability of itraconazole and voriconazole. Cardiac toxicities with itraconazole use particularly require ardent monitoring of the patients to avoid adverse events like heart failure, which predisposes to increased mortality. Randomised trials are encouraged to guide appropriate selection of a safe and efficacious first-line antifungals for the management of CPA.

## Supporting information

**S1 Fig. Funnel plot for voriconazole studies.**
(DOCX)

**S2 Fig. Sensitivity analysis forest plot for voriconazole.**
(PNG)

**S1 File. PRISMA checklist.**
(DOCX)

**S2 File. Search strategy.**
(DOCX)

**S3 File. Risk of bias assessment.**
(DOCX)

## Author Contributions

**Conceptualization:** Felix Bongomin.

**Data curation:** Joseph Baruch Baluku, Laura Russell, Felix Bongomin.

**Formal analysis:** Ronald Olum, Joseph Baruch Baluku, Felix Bongomin.

**Methodology:** Joseph Baruch Baluku, Felix Bongomin.

**Validation:** Joseph Baruch Baluku, Andrew Kazibwe, Felix Bongomin.

**Writing – original draft:** Ronald Olum, Joseph Baruch Baluku, Andrew Kazibwe, Laura Russell, Felix Bongomin.

**Writing – review & editing:** Ronald Olum, Joseph Baruch Baluku, Andrew Kazibwe, Laura Russell, Felix Bongomin.

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
