## [Decision Letter · Decision Letter 0]

3 Aug 2020

PONE-D-20-17016

Tolerability of Oral Itraconazole and Voriconazole for The Treatment of Chronic Pulmonary Aspergillosis: A Systematic Review and Meta-Analysis

PLOS ONE

Dear Dr. Bongomin,

Thank you for submitting your manuscript to PLOS ONE. After careful consideration, we feel that it has merit but does not fully meet PLOS ONE’s publication criteria as it currently stands. Therefore, we invite you to submit a revised version of the manuscript that addresses the points raised during the review process.

We look forward to receiving your revised manuscript.

Kind regards,

Ahmed Negida, MD

Academic Editor

PLOS ONE

Journal Requirements:

2. We note that 516 manuscripts were identified through database searching but that only 56 records (or only about 10% of records) remained after duplicates were removed. Please double check your duplicate removal strategy and confirm that the several hundred records removed were indeed duplicates.

3. Please provide the date(s) when the databases used in this study were last accessed.

4. Please report the heterogeneity analysis results across studies (assessed using Q statistics) and comment on the appropriateness of a meta-analysis.

5. Please amend your list of authors on the manuscript to ensure that each author is linked to an affiliation. Authors’ affiliations should reflect the institution where the work was done (if authors moved subsequently, you can also list the new affiliation stating “current affiliation:….” as necessary).

Reviewers' comments:

Reviewer's Responses to Questions

**Comments to the Author**

1. Is the manuscript technically sound, and do the data support the conclusions?

Reviewer #1: Yes

Reviewer #2: Yes

Reviewer #3: Partly

Reviewer #4: Yes

Reviewer #5: Partly

2. Has the statistical analysis been performed appropriately and rigorously? 

Reviewer #1: Yes

Reviewer #2: Yes

Reviewer #3: Yes

Reviewer #4: Yes

Reviewer #5: Yes

3. Have the authors made all data underlying the findings in their manuscript fully available?

Reviewer #1: Yes

Reviewer #2: Yes

Reviewer #3: Yes

Reviewer #4: Yes

Reviewer #5: Yes

4. Is the manuscript presented in an intelligible fashion and written in standard English?

Reviewer #1: Yes

Reviewer #2: Yes

Reviewer #3: No

Reviewer #4: Yes

Reviewer #5: Yes

5. Review Comments to the Author

Reviewer #1: This review aimed to to evaluate the frequency of adverse events of itraconazole and voriconazole for the treatment of chronic pulmonary aspergillosis. This is the first meta-analysis that discuss this point. I have some comments that will enhance the quality of this review:

General

1- The whole manuscript needs extensive revision for language mistakes.

Abstract

2- Sections of results and conclusions need to be paraphrased.

3- Mention all searched databases as reported in the methods.

Introduction

4- The third paragraph (lines 64 to 69) is very confusing; try to make it simple for reading.

5- The efficacy (response rate) of Itraconazole and voriconazole needs further references.

Methods

6- PubMed is a search engine not database (You searched MEDLINE through PubMed.

7- Cite the manifacure info of HDAS program; try to repeat this step (remove duplicates) with other program such as Mendeley or EndNote as I have some doubts with the results of HDAS program (It removed more than 90% of the downloaded citations!!!).

8- Write the initials of the reviewers involved in the data extraction process

9- Which random model you used in your analysis?

Results

10- Add a section of the findings of the potential source of bias assessment in the results.

11- Add a forest plot figure for the sensitivity analysis of Voriconazole in the supplementary file

Reviewer #2: This is an interesting study and the authors compared the safety, efficacy, and risks of Oral Itraconazole and Voriconazole in the treatment of chronic pulmonary aspergillosis. The paper is generally well written and structure. The quality of the manuscript is scientifically sound. Chronic pulmonary aspergillosis requires extensive and long-term treatment with anti-fungal medications. Adverse events are very common with the use of those medications. Many systematic reviews and metaanlyses investigated each of the treatment but none compared the efficacy of the both drugs in the treatment of chronic pulmonary aspergillosis. The authors found that most of the adverse events were observed in about one-fourth of patients treated with itraconazole and just over two-third of patients treated with voriconazole. This information can be used to educate patients prior to commencement of these antifungal therapies but also to guide the future clinical trials to investigate both treatment together. I think that systematic review will serve as an additional to the management of chronic pulmonary aspergillosis.

Reviewer #3: The authors performed a systematic review to evaluate the frequency of adverse events of itraconazole and voriconazole for the treatment of chronic pulmonary aspergillosis. They concluded that AEs were observed in about one-fourth of patients treated with itraconazole, and just over two-thirds of patients treated with voriconazole. I have few concerns:

Abstract

1) Methods

- It is a random-effects model, not “Random effect”

- Also, the databases searched are inconsistent with the manuscript text, please recheck

2) Results: No need for I2 and P values.

Methods

* Search strategy

- Were MESH terms used?

- The authors mentioned, “The search was limited to human studies and only studies written in English were selected”, were databases filters used to choose these studies?

- Did not the authors check the abstracts of articles not in English? Some relevant studies may have been missed.

- Why did not the authors assess the efficacy of both drugs, along with assessing the safety outcomes? (I.e., Overall, Clinical, Radiological, responses ... etc.)

- Line 118: replace “Qualitative assessment” with the risk of bias assessment

- Line 133: It is a random-effects model, not “Random effect”

Results

- Line 142: “A total of 10 studies”, add references for the included studies.

- Line 144: “Of the 10 studies, 7 were retrospective [4, 5, 12-15]”, these citations are for 6 studies, not 7.

- Add a paragraph about the results of the “risk of bias assessment.”

- Please interpret the funnel plot.

- Figure 1: How is it possible that records retrieved after the removal of duplicates (n = 56), and the records screened (n = 21) !! please recheck.

Discussion

- I would recommend that basic information about the antifungal therapies should be moved to the introduction.

Language: The entire manuscript needs extensive professional revision for grammatical errors and stylistic editing to improve the quality of English. For example:

Line 33: “case reports and case series”, consider inserting a comma before “and”

Line 46: “compared tolerability”, an article is missing before the word tolerability

Line 50: two-thirds not “two-third”

Line 247: “is varies” the verb “varies” does not work with “is” in this sentence

Line 260: “in the treatment of with CPA”, incorrect preposition use, etc.

Reviewer #4: Reviewer report on the manuscript PONE-D-20-17016_titled " Tolerability of Oral Itraconazole and Voriconazole for The Treatment of Chronic Pulmonary Aspergillosis: A Systematic Review and Meta-Analysis". The present research was aimed to evaluate the frequency of adverse events of itraconazole and voriconazole for the treatment of CPA. I have some major limitations that that should be considered for considering publication of this manuscript, these include:

1. In the abstract

a. (line 37-38) the sentence started with" We included 10 eligible studies, in 8 studies, 366 patients were treated with voriconazole (69%) and 168 with itraconazole (31%) in 2 studies" was ended with 2 studies???.. it should be rephrased to be understandable. I understand that, the 366 patients were enrolled in the 8 studies but what about the two other studies???? What were the differences between the 8 and the 2 studies?

b. Lines 40 and 42, the authors should include the number of patients between brackets adjacent to the 36% and 25%.

c. Lines 42 and 43, the author should not start the sentence with numbers and both sentences should sentences should be lined to avoid repetition of verbs.

d. The whole abstract should be rephrased particularly the % and (number of patients) in order to presented in more clearly and understandable way.

2. In the introduction

a. "line 77" the sentence started with "Few studies have reported adverse events …….etc) however, the authors did not make citation, therefore, the author should inserted the appropriate citation for these few studies.

b. Abbreviations should be defined at first mention and used consistently thereafter. In line 77, the author should replace adverse events with the abbreviation (AEs) as previously indicated.

c. There are a lot of relevant literature are available and I recommend the authors to be include them in both introduction and discussion sections, examples

i. Bongomin F, Asio LG, Baluku JB, Kwizera R, Denning DW. Chronic Pulmonary Aspergillosis: Notes for a Clinician in a Resource-Limited Setting Where There Is No Mycologist. J Fungi (Basel). 2020;6(2):75. Published 2020 Jun 2. doi:10.3390/jof6020075.

ii. Jenks JD, Hoenigl M. Treatment of Aspergillosis. J Fungi (Basel). 2018;4(3):98. Published 2018 Aug 19. doi:10.3390/jof4030098

iii. Alastruey-Izquierdo A, Cadranel J, Flick H, et al. Treatment of Chronic Pulmonary Aspergillosis: Current Standards and Future Perspectives. Respiration. 2018;96(2):159-170. doi:10.1159/000489474

iv. Maghrabi F, Denning DW. The Management of Chronic Pulmonary Aspergillosis: The UK National Aspergillosis Centre Approach. Curr Fungal Infect Rep. 2017;11(4):242-251. doi:10.1007/s12281-017-0304-7

v. Baxter CG, Marshall A, Roberts M, Felton TW, Denning DW. Peripheral neuropathy in patients on long-term triazole antifungal therapy. J Antimicrob Chemother. 2011;66(9):2136-2139. doi:10.1093/jac/dkr233

vi. Sambatakou H, Dupont B, Lode H, Denning DW. Voriconazole treatment for subacute invasive and chronic pulmonary aspergillosis. Am J Med. 2006;119(6):527.e17-527.e5.27E24. doi:10.1016/j.amjmed.2005.11.028

3. Type of article should be changed from" Research article" to "review article) since it was meta-analysis and in silico study

Therefore, and according to the above mentioned remarks I advised minor revision of this manuscript under its current status.

Reviewer #5: - Line 13: This affiliation is missing at the author's list

- The number of studies included are considered to be small. Eight studies and some of them are providing data for one of the two drugs only. This affects the estimation of variation between studies.

- I suggest the authors to be more focused on serious AEs that necessitate the discontinuation of treatment.

- What is the explanation of having more patient treated with voriconazole than those treated with itraconazole (Results line 38) although the second is considered to be the first line.

- In conclusion section, the authors are suggesting the use of itraconazole over the use of voriconazole while it is already known for itraconazole to be the first line of treatment.

- Line 323: which, which

6. PLOS authors have the option to publish the peer review history of their article (what does this mean?). If published, this will include your full peer review and any attached files.

Reviewer #1: No

Reviewer #2: No

Reviewer #3: No

Reviewer #4: No

Reviewer #5: No

---

## [Author Response · Author response to Decision Letter 0]

11 Aug 2020

Authors’ Responses

Editor

Journal Requirements:

 Authors’ Response:

The manuscript has been re-formatted according to PLOS Ones guidelines. Thank you.

2. We note that 516 manuscripts were identified through database searching but that only 56 records (or only about 10% of records) remained after duplicates were removed. Please double check your duplicate removal strategy and confirm that the several hundred records removed were indeed duplicates.

 Authors’ Response:

We apologise for this error in data extraction from the HDSA output. We had only 17 results from embase and 7 from Medline, and 5 additional from searching references of the selected articles. The Prisma flow diagram and the highlighted search strategy are provided.

3. Please provide the date(s) when the databases used in this study were last accessed.

Authors’ Response: 

The databases were accessed on the May 12th, 2020 as indicated in the search strategy file.

4. Please report the heterogeneity analysis results across studies (assessed using Q statistics) and comment on the appropriateness of a meta-analysis.

Authors’ Response: 

Heterogeneity was assessed using both Q statistics (chi-square = 86.88, p<0.001 and I2 statistics (i2 = 91.94, p<0.001). However, only the I2 statistics was reported in the manuscript since it is superior to Q statistics especially in meta-analyses with fewer studies. The interpretation and implication of the notable heterogeneity has been discussed as a limitation in the discussion section.

Authors’ Response: 

5. Please amend your list of authors on the manuscript to ensure that each author is linked to an affiliation. Authors’ affiliations should reflect the institution where the work was done (if authors moved subsequently, you can also list the new affiliation stating “current affiliation:….” as necessary).

Authors’ Response: 

Thank you. The authors’ list and affiliations have been amended appropriately.

Authors’ Response:

Thank you. The captions have been added according to the journal guidelines.

Reviewer #1: This review aimed to to evaluate the frequency of adverse events of itraconazole and voriconazole for the treatment of chronic pulmonary aspergillosis. This is the first meta-analysis that discuss this point. I have some comments that will enhance the quality of this review:

General

1- The whole manuscript needs extensive revision for language mistakes.

Authors’ response:

Thank you. The manuscript was sent to a native English speaker for proof-reading and grammatical editing. We believe the paper can now be understood clearly.

Abstract

2- Sections of results and conclusions need to be paraphrased.

Authors’ response: Thank you. We may have revised these sections

3- Mention all searched databases as reported in the methods.

Authors’ response:

Thank you. All the searched databases reported in the methodology have been indicated as suggested.

Introduction

4- The third paragraph (lines 64 to 69) is very confusing; try to make it simple for reading.

Authors’ response: Thank you, this paragraph has been revised.

5- The efficacy (response rate) of Itraconazole and voriconazole needs further references.

Authors’ response: Many thanks, this has been amended.

Methods

6- PubMed is a search engine not database (You searched MEDLINE through PubMed.

Authors’ response:

Thank you. This has been rectified as suggested.

7- Cite the manufacture info of HDAS program; try to repeat this step (remove duplicates) with other program such as Mendeley or EndNote as I have some doubts with the results of HDAS program (It removed more than 90% of the downloaded citations!!!).

Authors’ response:

The manufacture of HDAS program has been duly cited. 

8- Write the initials of the reviewers involved in the data extraction process

Authors’ response:

Thank you. The initials o the reviewers have been included in the main text as suggested.

9- Which random model you used in your analysis?

Authors’ response:

A random-effects model was used for meta-analysis. The metaprop command used (Command: metaprop n N, random) specifies a random-effects model using the method of DerSimonian and Laird, with the estimate of heterogeneity being taken from the inverse-variance fixed-effect model (Nyaga VN, Arbyn M, Aerts M. Metaprop: a Stata command to perform meta-analysis of binomial data. Archives of Public Health. 2014 Dec 1;72(1):39.).

Results

10- Add a section of the findings of the potential source of bias assessment in the results.

Authors’ response:

A paragraph on the risk of bias assessment has been added in the results section as suggested. Thank you.

11- Add a forest plot figure for the sensitivity analysis of Voriconazole in the supplementary file

Authors’ response:

Thank you. The forest plot for the sensitivity analysis has been submitted as a supplementary file and cited in-text.

Reviewer #2: This is an interesting study and the authors compared the safety, efficacy, and risks of Oral Itraconazole and Voriconazole in the treatment of chronic pulmonary aspergillosis. The paper is generally well written and structure. The quality of the manuscript is scientifically sound. Chronic pulmonary aspergillosis requires extensive and long-term treatment with anti-fungal medications. Adverse events are very common with the use of those medications. Many systematic reviews and metanalyses investigated each of the treatment but none compared the efficacy of the both drugs in the treatment of chronic pulmonary aspergillosis. The authors found that most of the adverse events were observed in about one-fourth of patients treated with itraconazole and just over two-third of patients treated with voriconazole. This information can be used to educate patients prior to commencement of these antifungal therapies but also to guide the future clinical trials to investigate both treatments together. I think that systematic review will serve as an additional to the management of chronic pulmonary aspergillosis.

Authors’ response:

Thank you so much for the kind words.

Reviewer #3: The authors performed a systematic review to evaluate the frequency of adverse events of itraconazole and voriconazole for the treatment of chronic pulmonary aspergillosis. They concluded that AEs were observed in about one-fourth of patients treated with itraconazole, and just over two-thirds of patients treated with voriconazole. I have few concerns:

Abstract

1) Methods

- It is a random-effects model, not “Random effect”

- Also, the databases searched are inconsistent with the manuscript text, please recheck

Authors’ response:

Thank you. The suggestions have been accepted and changed accordingly in the abstract.

2) Results: No need for I2 and P values.

Authors’ response:

Thank you. I2 and P values have been eliminated from the abstract as suggested.

Methods

* Search strategy

- Were MESH terms used?

Authors’ response:

The full search strategy used was ((((ITRACONAZOLE/ OR (Itraconazole).ti,ab OR (voriconazole).ti,ab OR VORICONAZOLE/) AND ("chronic pulmonary aspergillosis" OR "chronic cavitary aspergillosis" OR "chronic fibrosing pulmonary aspergillosis" OR "simple aspergilloma").ti,ab) AND ((adverse ADJ3 (reaction* OR effect OR affect)).ti,ab OR (toxic*).ti,ab OR "DRUG-RELATED SIDE EFFECTS AND ADVERSE REACTIONS"/ OR ("side effect*").ti,ab)) NOT ("invasive aspergillosis" OR "allergic aspergillosis").ti,ab. The strategy used Medline MESH terms where possible, in this case for Intraconazole, Voriconazole and Drug related side effects and adverse reactions.

- The authors mentioned, “The search was limited to human studies and only studies written in English were selected”, were databases filters used to choose these studies?

Authors’ response:

Database filters were used to limit to English language papers and human only studies. 

- Did not the authors check the abstracts of articles not in English? Some relevant studies may have been missed.

Authors’ response: We limited our search strategy only to published English literature. 

- Why did not the authors assess the efficacy of both drugs, along with assessing the safety outcomes? (I.e., Overall, Clinical, Radiological, responses ... etc.)

Authors’ response: Many thanks for these suggestions. We sought to answer a single hypothesis on safety. However, efficacy studies would be a good subject for future meta-analysis. 

- Line 118: replace “Qualitative assessment” with the risk of bias assessment

Authors’ response:

Thank you. The suggestion has been accepted and change accordingly.

- Line 133: It is a random-effects model, not “Random effect”

Authors’ response:

Thank you. The suggestion has been accepted and changed accordingly.

Results

- Line 142: “A total of 10 studies”, add references for the included studies.

- Line 144: “Of the 10 studies, 7 were retrospective [4, 5, 12-15]”, these citations are for 6 studies, not 7.

Authors’ response:

Thank you for this observation. The references have been added as suggested. There was an obvious error in counting the number of eligible studies that we realised immediately after submission. The study by Bongomin et al , 2018 reported adverse events for both itraconazole and voriconazole. This was unfortunately counted as two studies. A total of 9 studies were eligible instead. The error has been corrected throughout the manuscript. Therefore, only 6 studies were retrospective. 

- Add a paragraph about the results of the “risk of bias assessment.”

Authors’ response:

Thank you. A paragraph has been added as suggested. 

- Please interpret the funnel plot.

Authors’ response:

Thank you. The funnel plot has been interpreted.

- Figure 1: How is it possible that records retrieved after the removal of duplicates (n = 56), and the records screened (n = 21) !! please recheck.

Authors’ response: Many thanks for spotting this. This was an error in extraction of HDAS output. We had only 17 results from embase and 7 from Medline, and 5 additional from searching references of the selected articles. The Prisma flow diagram and the highlighted search strategy are provided.

Discussion

- I would recommend that basic information about the antifungal therapies should be moved to the introduction.

Authors’ response:

Thanks for this suggestion. However, we thought the discussion section was a good place to explain the pharmacology of these agents. So We have left it as is .

Language: The entire manuscript needs extensive professional revision for grammatical errors and stylistic editing to improve the quality of English. For example:

Line 33: “case reports and case series”, consider inserting a comma before “and”

Line 46: “compared tolerability”, an article is missing before the word tolerability

Line 50: two-thirds not “two-third”

Line 247: “is varies” the verb “varies” does not work with “is” in this sentence

Line 260: “in the treatment of with CPA”, incorrect preposition use, etc.

Authors’ response:

Thank you for the concerns. The above outlined errors have been corrected as suggested. The manuscript was also sent to a native English speaker to make grammatical corrections on the whole paper. We believe the science in the paper can now be understood clearly by the readers.

Reviewer #4: Reviewer report on the manuscript PONE-D-20-17016_titled " Tolerability of Oral Itraconazole and Voriconazole for The Treatment of Chronic Pulmonary Aspergillosis: A Systematic Review and Meta-Analysis". The present research was aimed to evaluate the frequency of adverse events of itraconazole and voriconazole for the treatment of CPA. I have some major limitations that that should be considered for considering publication of this manuscript, these include:

1. In the abstract

a. (line 37-38) the sentence started with" We included 10 eligible studies, in 8 studies, 366 patients were treated with voriconazole (69%) and 168 with itraconazole (31%) in 2 studies" was ended with 2 studies??? it should be rephrased to be understandable. I understand that, the 366 patients were enrolled in the 8 studies but what about the two other studies???? What were the differences between the 8 and the 2 studies?

Authors’ response:

Thank you. The statements have been rephrased and is now clear. In all the eligible studies included, a combined total of 534 patients were enrolled. 366 (69%) were treated with voriconazole and 168 (31%) with itraconazole. 8 studies reported patients treated with voriconazole and 2 studies reported those treated with itraconazole.

b. Lines 40 and 42, the authors should include the number of patients between brackets adjacent to the 36% and 25%.

Authors’ response:

Thank you. This has been included as suggested.

c. Lines 42 and 43, the author should not start the sentence with numbers and both sentences should sentences should be lined to avoid repetition of verbs.

Authors’ response:

Thank you. The two sentences have been corrected as suggested by the reviewer. 

d. The whole abstract should be rephrased particularly the % and (number of patients) in order to presented in more clearly and understandable way.

Authors’ response:

The entire abstract has been rephrased to present the results more clearly and in a comprehensible manner. Thank you.

2. In the introduction

a. "line 77" the sentence started with "Few studies have reported adverse events …etc) however, the authors did not make citation, therefore, the author should insert the appropriate citation for these few studies.

Authors’ response:

Thank you for this important observation. Citations supporting the statement have been added.

b. Abbreviations should be defined at first mention and used consistently thereafter. In line 77, the author should replace adverse events with the abbreviation (AEs) as previously indicated.

Authors’ response:

Thank you. This has been rectified.

c. There are a lot of relevant literature are available and I recommend the authors to be include them in both introduction and discussion sections, examples

i. Bongomin F, Asio LG, Baluku JB, Kwizera R, Denning DW. Chronic Pulmonary Aspergillosis: Notes for a Clinician in a Resource-Limited Setting Where There Is No Mycologist. J Fungi (Basel). 2020;6(2):75. Published 2020 Jun 2. doi:10.3390/jof6020075.

ii. Jenks JD, Hoenigl M. Treatment of Aspergillosis. J Fungi (Basel). 2018;4(3):98. Published 2018 Aug 19. doi:10.3390/jof4030098

iii. Alastruey-Izquierdo A, Cadranel J, Flick H, et al. Treatment of Chronic Pulmonary Aspergillosis: Current Standards and Future Perspectives. Respiration. 2018;96(2):159-170. doi:10.1159/000489474

iv. Maghrabi F, Denning DW. The Management of Chronic Pulmonary Aspergillosis: The UK National Aspergillosis Centre Approach. Curr Fungal Infect Rep. 2017;11(4):242-251. doi:10.1007/s12281-017-0304-7

v. Baxter CG, Marshall A, Roberts M, Felton TW, Denning DW. Peripheral neuropathy in patients on long-term triazole antifungal therapy. J Antimicrob Chemother. 2011;66(9):2136-2139. doi:10.1093/jac/dkr233

vi. Sambatakou H, Dupont B, Lode H, Denning DW. Voriconazole treatment for subacute invasive and chronic pulmonary aspergillosis. Am J Med. 2006;119(6):527.e17-527.e5.27E24. doi:10.1016/j.amjmed.2005.11.028

Authors’ response:

Thank you for the valuable suggestions. We have reviewed the above papers and found them excellent and relevant for our paper. They have been included appropriately.

3. Type of article should be changed from" Research article" to "review article) since it was meta-analysis and in silico study

Authors’ response:

Thanks for this suggestions. However, PLoS one considers SR and MA as Original Articles

Reviewer #5: 

- Line 13: This affiliation is missing at the author's list.

The affiliation has been updated in the authors’ list. Thank you for the keen observation.

- The number of studies included are considered to be small. Eight studies and some of them are providing data for one of the two drugs only. This affects the estimation of variation between studies.

Authors’ response: We agree with the reviewer. We acknowledged that most studies reported on AEs to a single agent and we also acknowledged that there is no published study comparing itra and vori head to head. Bongomin et al 2018 is the only study that reported the use of itra and vori in more than 150 patients. Again this was an observational study. We recommened that more studies are required to provide definitive answers. Thank you

- I suggest the authors to be more focused on serious AEs that necessitate the discontinuation of treatment.

Authors’ response: Many thanks for this. We presented both sides of the data, overall AEs and AEs warranting discontinuation of therapy. We believe both parts of the story is useful to the patients.

- What is the explanation of having more patient treated with voriconazole than those treated with itraconazole (Results line 38) although the second is considered to be the first line.

Authors’ response: this finding also surprised us. We know that the recommendation of itra as first line is based mainly on expert opinions and a single RCT which had less than 20 patients on itra. And the other possible explanation is publication bias, where more studies on Vori are published compared to itra.

- In conclusion section, the authors are suggesting the use of itraconazole over the use of voriconazole while it is already known for itraconazole to be the first line of treatment.

Authors’ response: Thanks for this observation, however, we said “more tolerable” and we didn’t suggest it as a preferred first line. Current guidelines suggest Itra and vori are alternative firstlline agents and the choice of use of itra or vori is based on the clinicians assessment of risks and benefits for each individual patient. 

- Line 323: which, which

Authors’ response:

Thank you. This has been rectified accordingly.

---

## [Decision Letter · Decision Letter 1]

14 Sep 2020

PONE-D-20-17016R1

Tolerability of oral itraconazole and voriconazole for the treatment of chronic pulmonary aspergillosis: a systematic review and meta-analysis

PLOS ONE

Dear Dr. Bongomin,

Thank you for submitting your manuscript to PLOS ONE. After careful consideration, we feel that it has merit but does not fully meet PLOS ONE’s publication criteria as it currently stands. Therefore, we invite you to submit a revised version of the manuscript that addresses the points raised during the review process.

ACADEMIC EDITOR

The article is acceptable after addressing our minor comments "shown below".

We look forward to receiving your revised manuscript.

Kind regards,

Ahmed Negida, MD

Academic Editor

PLOS ONE

Additional Editor Comments (if provided):

1- Please, address the comments of Reviewer 1.

2- I suggest adding publication bias assessment & the funnel plot in the online supplementary files not in the main manuscript. According to Egger et al. the assessment of publication bias in fewer than 10 studies is not statistically reliable. While you information is indicative and informative of the possibility of publication bias, it is not a firm reliable method to judge the current evidence (based on the limited number of analyzed studies). Therefore, I would suggest removing this part to an online supplementary file.

Reviewers' comments:

Reviewer's Responses to Questions

**Comments to the Author**

1. If the authors have adequately addressed your comments raised in a previous round of review and you feel that this manuscript is now acceptable for publication, you may indicate that here to bypass the “Comments to the Author” section, enter your conflict of interest statement in the “Confidential to Editor” section, and submit your "Accept" recommendation.

Reviewer #1: All comments have been addressed

Reviewer #2: All comments have been addressed

Reviewer #4: All comments have been addressed

2. Is the manuscript technically sound, and do the data support the conclusions?

Reviewer #1: Yes

Reviewer #2: Yes

Reviewer #4: (No Response)

3. Has the statistical analysis been performed appropriately and rigorously? 

Reviewer #1: Yes

Reviewer #2: Yes

Reviewer #4: Yes

4. Have the authors made all data underlying the findings in their manuscript fully available?

Reviewer #1: Yes

Reviewer #2: Yes

Reviewer #4: Yes

5. Is the manuscript presented in an intelligible fashion and written in standard English?

Reviewer #1: Yes

Reviewer #2: Yes

Reviewer #4: Yes

6. Review Comments to the Author

Reviewer #1: The authors addressed all my comments, and I think that the manuscript now is well-written and well-presented; however, I have some concerns about the searching process:

Why you did you exclude the Web of Science and Cochrane library?

Did you repeated the searching and screening process?

How could the number of articles in EMBASE decreased from 516 to 17? Did you applied any filters?

Reviewer #2: The authors addressed all the required comments. Therefore, I think that that the manuscript is suitable for review at PLOS ONE.

Reviewer #4: The review comments have been addressed by authors. all required questions have been answered by authors and the responses meet formatting specification

7. PLOS authors have the option to publish the peer review history of their article (what does this mean?). If published, this will include your full peer review and any attached files.

Reviewer #1: No

Reviewer #2: No

Reviewer #4: No

---

## [Author Response · Author response to Decision Letter 1]

18 Sep 2020

Response to Editor 

1- Please, address the comments of Reviewer 1.

Authors’ response: Thank you editor. We have addressed the reviewer 1’s comments below.

2- 

2- I suggest adding publication bias assessment & the funnel plot in the online supplementary files not in the main manuscript. According to Egger et al. the assessment of publication bias in fewer than 10 studies is not statistically reliable. While you information is indicative and informative of the possibility of publication bias, it is not a firm reliable method to judge the current evidence (based on the limited number of analyzed studies). Therefore, I would suggest removing this part to an online supplementary file

Authors’ response: Thank you for these insightful suggestions. The bias assessment and the funnel plots have been removed from the main manuscript and transferred to supplementary materials. 

Response to reviewer 1

Reviewer #1: The authors addressed all my comments, and I think that the manuscript now is well-written and well-presented; however, I have some concerns about the searching process:

Why you did you exclude the Web of Science and Cochrane library?

Authors’ response: We thank the reviewer for this query: Our specialist medical librarian (LR – an author to this paper) conducted an extensive search inclusive of both WoS and Cochrane library. Howver, she retrieved very few articles from this two databases – all of which were already picked up by Embase and Medline – Therefore WoS and Cochrane library did not contribute any unique search and so we did not include them on the PRISMA flow diagram

Did you repeated the searching and screening process?

Authors’ response: Yes, we did and we got the same output – no new articles were identified. We further confirmed our search output as outlined in Figure 1.

How could the number of articles in EMBASE decreased from 516 to 17? Did you applied any filters?

Authors’ response: Many thanks for this confirmatory question again. As we responded to your comments in the first review, this was an error in extraction of HDAS output. We had only 17 results from embase and 7 from Medline (we have provided the search output as a supplementary material), and 5 additional articles from searching references of the selected articles. Yes, we applied filters – a qualified Medical Librarian did these searches.

Thank you very much for improving the quality of this paper. Much appreciated.

---

## [Editor Report · Decision Letter 2]

25 Sep 2020

Tolerability of oral itraconazole and voriconazole for the treatment of chronic pulmonary aspergillosis: a systematic review and meta-analysis

PONE-D-20-17016R2

Dear Dr. Bongomin,

We’re pleased to inform you that your manuscript has been judged scientifically suitable for publication and will be formally accepted for publication once it meets all outstanding technical requirements.

Kind regards,

Ahmed Negida, MD

Academic Editor

PLOS ONE

---

## [Editor Report · Acceptance letter]

28 Sep 2020

PONE-D-20-17016R2 

Tolerability of oral itraconazole and voriconazole for the treatment of chronic pulmonary aspergillosis: a systematic review and meta-analysis 

Dear Dr. Bongomin:

I'm pleased to inform you that your manuscript has been deemed suitable for publication in PLOS ONE. Congratulations! Your manuscript is now with our production department. 

Kind regards, 

on behalf of

Dr. Ahmed Negida 

Academic Editor

PLOS ONE